# A Progressive Skip Reasoning Fusion Method for Multi-Modal Classification

## ABSTRACT

In multi-modal classification tasks, a good fusion algorithm can effectively integrate and process multi-modal data, thereby significantly improving its performance. Researchers often focus on the design of complex fusion operators and have proposed numerous fusion operators, while paying less attention to the design of feature fusion usage, specifically how features should be fused to better facilitate multi-modal classification tasks. In this article, we propose a progressive skip reasoning fusion network (PSRFN) to make some attempts to address this issue. Firstly, unlike most existing multi-modal fusion methods that only use one fusion operator in a single stage to fuse all view features, PSRFN utilizes the progressive skip reasoning (PSR) block to fuse all views with a fusion operator at each layer. Specifically, each PSR block utilizes all view features and the fused features from the previous layer to jointly obtain the fused features for the current layer. Secondly, each PSR block utilizes a dual-weighted fusion strategy with learnable parameters to adaptively allocate weights during the fusion process. The first level of weighting assigns weights to each view feature, while the second level assigns weights to the fused features from the previous layer and the fused features obtained from the first level of weighting in the current layer. This strategy ensures that the PSR block can dynamically adjust the weights based on the actual contribution of features. Finally, to enable the model to fully utilize feature information from different levels for feature fusion, the skip connections are adopted between PSR blocks employing them. Extensive experiment results on six real multi-modal datasets show that a better usage for fusion operator is indeed able to improve performance.

## CCS CONCEPTS

• **Multimedia Content Understanding** → **Multimodal Fusion**.

## KEYWORDS

Multi-modal fusion, Classification, Progressive fusion

## 1 INTRODUCTION

With rapid development of multimedia and representation learning methods, data are generally represented with multiple group of features from different views [12, 19, 28]. We refer to this type of data as multi-view data, and each group of features is termed one view (it is noted that like other works [9, 22] the terms "view" and "modality" are used interchangeably in this paper). For example, we can extract color, shape and texture features from images, multiple kinds of language features from texts, and acoustics and energy features from audio. Multi-modal data have facilitated the development of many tasks such as classification [16, 20, 34], clustering [35, 36, 40], and feature selection [6, 14].

Among these multi-modal tasks, multi-modal classification technique attracts more and more attention of researchers from academics and industry with increasing demands for multimedia data management. Many studies show an effective fusion operator is very important for such task [3]. Hence the researchers in this community pay enormous attention to the fusion operator design. For example, Liang et al. [22] introduced the association information between modality features into multi-modal data fusion and proposed an association-based fusion strategy for multi-modal classification (MMC) in an interpretable manner. Hu et al. [11] proposed a fuzzy fusion method and this method used a fuzzy operator to fuse all kernel matrixes that are obtained by operating different kernel functions on each view. Liu et al. [26] developed an efficient low-rank multi-modal fusion method with modality-specific factors to address the problem of tensor-based methods suffering from exponential increase in dimensions and in computational complexity. Hou et al. [10] aimed to address the problem of existing tensor-based methods ignoring the complex local intercorrelations and proposed a polynomial tensor pooling method (PTP) for integrating multi-view features by considering high-order moment. So far, a huge number of fusion operators [2, 29, 37] have been proposed. However, fusion operator usage is rarely studied.

Moreover, most existing fusion methods only conduct once fusion for learned multi-view representation, and then the fused features are passed to a classifier for decision-making. The fused features obtained with the strategy may be sub-optimal due to insufficient interaction among views. One may ask one question that "*Can the fused features be enhanced via multiple interactions with multi-view representation*"?

Many applications such as face recognition and speaker recognition benefit from the hierarchical features of deep learning. This suggests that the original features could not be the best fit for the current task and one or more feature transformations (namely representation learning [4]) are needed for better performance. Similarly, hierarchical features for multi-view features are also being learned for performance gain. In this case, multi-view fusion faces one issue: *which hierarchical features should the fusion operator operate on ?* Moreover, in representation learning, to learn an effective feature representation, we always transform features using a layer-by-layer manner. Based on these observations, it is reasonable that fusion should occur on each layer and the fusion representation should be passed layer by layer.

Accordingly, the article studies the issue of *the fusion operator usage in hierarchical multi-view features*. Specially, we propose a

*ACM MM, 2024, Melbourne, Australia*
© 2024 Copyright held by the owner/author(s). Publication rights licensed to ACM.
ACM ISBN 978-x-xxxx-xxxx-x/YY/MM
https://doi.org/10.1145/nnnnnnn.nnnnnnn

progressive skip reasoning fusion network (PSRFN). In PSRFN, views are transformed once, fusion will occur once. This process is performed by stacking multiple PSR blocks, and each PSR block is able to be armed with existing fusion operators. The PSR block takes view feature vectors at the current layer and the fused feature vector at the previous layer as inputs, fuse them, and then output the fused feature vector of the current layer.

Our main contributions are summarized as follows:

(1) A progressive fusion strategy is proposed, which incorporates the fused feature from the previous layer and the different view features from the current layeras the input for the current fusion layer, thus achieving progressive fusion. This strategy fully utilizes the complementarity of information from all views and gradually improves the discriminative ability of features. With the strategy, a progressive skip reasoning (PSR) block is proposed, which is easy to be modified and enhanced by replacing its fusion operation with existing fusion operators according to different tasks or datasets.

(2) A dual-weighted fusion strategy with learnable parameters is proposed, which can automatically adjust weights based on the actual contribution of each view feature and fusion layer feature. This strategy divides the weighted fusion process into two steps to achieve precise control of weights and fully utilize information from all views.

(3) A progressive skip reasoning fusion network (PSRFN) for multi-modal classification is designed by stacking multiple PSR Blocks. PSR is able to fuse different view features more than times according to the number of PSR blocks given by users. This way avoids the problem that users have to determine where views should be fused. Moreover, the information transmission between PSR blocks adopts skip connections, enabling the model to fully capture both deep and shallow features in multi-modal data, thereby enhancing the model's ability to understand the data. Additionally, skip connections also help alleviate the problem of gradient vanishing, allowing the model to learn feature representations more effectively during the training process.

(4) Extensive experiments conducted on six real multi-modal datasets verify the effectiveness of PSRFN.

## 2 RELATED WORK

### 2.1 Multi-Modal Classification

Multi-modal classification (MMC) aims to achieve a better and robust classification performance by integrating multiple features. Formally, let $\mathcal{X} = \mathbb{R}^{m_1} \times \mathbb{R}^{m_2} \times \cdots \times \mathbb{R}^{m_P}$ denote the instance space (or feature space) of $P$ modality representation, where $m_p(1 \le p \le P)$ is the feature dimension of $p$-th modality and $\mathcal{Y} = \{l_1, l_2, \cdots, l_q\}$ denote the label space with $q$ class labels. Denote $\mathcal{D}$ as an unknown distribution over $\mathcal{X} \times \mathcal{Y}$. A training set $D = \{(\boldsymbol{x}_i^p, y_i) | 1 \le p \le P, 1 \le i \le n\} \in (\mathcal{X} \times \mathcal{Y})^n$ is drawn identically and independently according to $\mathcal{D}$, where $\boldsymbol{x}_i^p = (x_{i1}^p, x_{i2}^p, \cdots, x_{im_p}^p) \in \mathbb{R}^{m_p}$ is the $p$-th modality feature vector with dimension $m_p$ and $y_i \in \mathcal{Y}$ is the known label associated with $\boldsymbol{x}_i^p$. The task of MMC is to learn a predictive function $f : \mathcal{X} \mapsto \mathcal{Y}$ from $D$ which can assign proper label $f(\boldsymbol{x}) \in \mathcal{Y}$ for unseen instance $\boldsymbol{x}$.

A multi-modal classification learner can be denoted as a two-tuple $\mathfrak{L} = (h, \mathcal{F})$, where $h$ is a learned decision function also called a classifier; $\mathcal{F}$ is a fusion function and it generally takes all view features and outputs their fused vector as the input of $h$.

### 2.2 View-Weighting Methods

The quality among multi-view features is different, lots of works consider their contribution to the final tasks. Multi-modal classification methods can be roughly divided into the feature level fusion-based (also namely early fusion) and decision level fusion-based (also namely late fusion). Based on where the view contribution are considered, these methods can be grouped into *feature* level weighting-based method (FW) and *decision* level weighting-based method (DW). FW learns the contribution weight of each feature [5, 15, 38, 41]. For example, EmbraceNet assigns 1 to the weight value of one view while 0 to others for each example according to a multinomial distribution. Yang et al. [38] proposed an adaptive-weighting discriminative regression approach (AWDR). AWDR adopts the square root form of view weight to distinguish features from different views. Zhang et al. [41] proposed discriminative multi-view fusion via adaptive regression (DMVF), it simultaneously discriminates the contribution diversity of different views and samples in an adaptive weighting manner, reducing the influence of low-quality views and outliers for classification. DW learns to assign weights at the decision level. For example, Han et al. [8] proposed a trusted multi-view classification (TMC), which models the confidence of each view at an evidence level using the Dempster-Shafer theory. Liu et al. [25] noticed that the fusion with Dempster's rule will produce counter-intuitive results, and adopted the opinion aggregation strategy to model the contribution degree of each view decision results.

### 2.3 Fusion Methods

Fusion function $\mathcal{F}$ plays a very important role in our PSR block. We will review related multi-modal classification methods from the perspective of the adopted fusion ways, then select some of them to design PSR block.

*2.3.1 Basic Fusion methods.* In the literatures of multi-modal learning, a common baseline for information fusion is to simply concatenate feature vectors across different modalities [32] such as images and texts, which is surprisingly performant in various multi-modal tasks. Other simple yet powerful methods include element-wise addition, multiplication and max-pooling [21]. Nonetheless, these element-wise operations are often limited in expressiveness as some noisy modalities could clutter the entire feature vectors.

*Concatenation:* We combine the information from both modalities using concatenation, i.e.,

$$o(x_i) = [U\text{MLP}(x_i^1), V\text{MLP}(x_i^2))] \tag{1}$$

where the function MLP is to map each view into a same size dimension, $[\cdot, \cdot]$ is the concatenation operator. The drawback of this fusion strategy is still considering the multi-view information as separate as each of them affects the classification independently.

*Addition:* We combine the information from both modalities using element-wise addition, i.e.,

$$o(x_i) = U\text{MLP}(x_i^1) + V\text{MLP}(x_i^2) \tag{2}$$

*Multiplication:* We combine the information from both modalities using element-wise product, i.e.,

$$o(x_i) = U\mathrm{MLP}(x_i^1) \times V\mathrm{MLP}(x_i^2) \tag{3}$$

*Max-pooling:* We combine the information from both modalities using element-wise maximization, i.e.,

$$o(x_i) = \max(U\mathrm{MLP}(x_i^1), V\mathrm{MLP}(x_i^2)) \tag{4}$$

where $(\cdot, \cdot)$ denotes concatenate two vectors.

According to the work [21], the addition has more chance to achieve a better performance and does not introduce extra learnable parameters. Hence, it is selected the representative of the five basic fusion operators to introduce to our PSR block.

*2.3.2 Advanced Fusion methods.* Recent bilinear models consider all pairwise interactions among given view features, providing richer expressive capacity than linear models. Recently, this technique has been successfully applied into egocentric hand action recognition [29], fine-grained visual recognition [23] and visual question answering (VQA) [17, 39]. In this section, we review two respective effective and efficient implementations of bilinear pooling.

Given two feature vectors $v_1$ and $v_2$ from two views, they can be fused with bilinear pooling as follows:

$$o_i = \sum_{j=1}^{N} \sum_{i=1}^{M} w_{ijk} x_j y_k + b_i = x_1^{\mathrm{T}} W_i x_2 + b_i \tag{5}$$

where $W_i \in \mathbb{R}^{m \times n}$ is a projection matrix, $c_i \in \mathbb{R}$ is the output of the bilinear model, and $b_i$ is a bias for the output $c_i$.

In the pooling processing, the bilinear method will yield a very high-dimensional fused vector and bring parameters of $L \times (N \times M + 1)$ including a bias vector $b$, where $L$ is the number of output features. It is not friendly fusion way for various computational resources. To overcome the problem of the curse of the parameters, Kim et al. proposed [17] the multi-view low-rank bilinear pooling (MLB) approach. First two input vectors (i.e., the image feature $x \in \mathbb{R}^m$ and the text feature $y \in \mathbb{R}^n$) are embedded into same dimension space with two low-rank projection matrices using two linear mappings without biases. Then, a joint representation is learnt with Hadamard product (element-wise multiplication) in a multiplicative way. Finally, a linear mapping with a bias is used to project the joint representations into an output vector for a given output dimension. To further increase model capacity, nonlinear activation like tanh is added after $c$. Since the MLB approach can generate feature vectors with low dimensions and deep networks with fewer model parameters, it has achieved very comparable performance to MCB. The process can be formalized as

$$c = \mathrm{MLB}(v_1, v_2) = U^{\mathrm{T}}(U_1^{\mathrm{T}} v_1 \circ U_2^{\mathrm{T}} v_2) + b \tag{6}$$

As pointed out by Yu et al. in [39], with the same dimensionality for the output features, MLB may suffer from insufficient representation due to approximating the outer product using a simple way that the features are directly projected to the low-dimensional output space and element-wise multiplication is performed. To overcome this problem, Yu et al. [39] proposed multimodal factorized bilinear pooling (MFB). First, the features from different modalities are expanded to a high-dimensional space and then integrated with

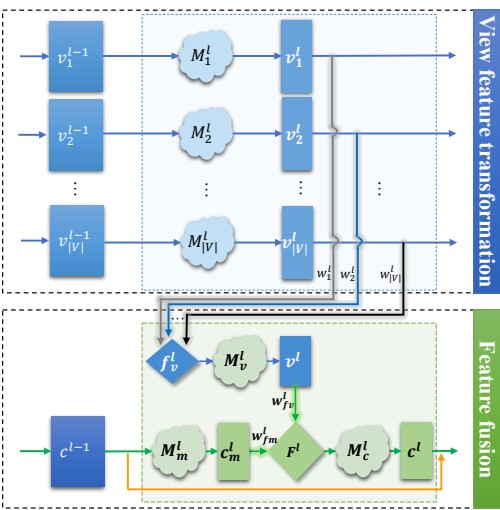

**Figure 1: One PSR block. It takes view features** $v_1^{l-1}, v_2^{l-1}, \cdots, v_{|V|}^{l-1}$ **as inputs, then transforms them to the transformed vectors** $v_1^l, v_2^l, \cdots, v_{|V|}^l$. **The transformed vectors and the fused vector** $c^{l-1}$ **result in the fused vector** $c^l$ **through the dual-weight fusion strategy.**

Hadamard product. After that, sum pooling followed by the normalization layers are performed to squeeze the high-dimensional feature into the compact output feature.

$$c = \mathrm{MFB}(v_1, v_2) = \mathrm{SumPool}(\hat{U}_1^{\mathrm{T}} v_1 \circ \hat{U}_2^{\mathrm{T}} v_2, k) \tag{7}$$

where the function $\mathrm{SunPool}(x, k)$ means using a one-dimensional non-overlapped window with the size $k$ to perform sum pooling over $x$.

It is clear from the formulas (5)-(7) that the key to advanced fusion operators lies in the outer product. Hence, the outer product is introduced to our PSR block.

## 3 PROGRESSIVE SKIP REASONING FUSION NETWORK

In this section, we first introduce a progressive skip reasoning (PSR) block, and then the progressive skip reasoning fusion network (PSRFN) for multi-modal classification is desinged by stacking multiple PSR blocks.

### 3.1 Progressive Skip Reasoning Block

As shown in Fig. 1, a PSR block consists of view feature transformation and feature fusion. It takes $|V|$ view feature vectors and the fusion vector from the previous layer, and then outputs the $|V|$ transformed view feature vectors and the fusion vector for the current layer.

**View feature transformation:** It aims to learn more hierarchical and richer feature representation for each original view features. It takes $|V|$ view feature vectors from the previous layer and outputs the $|V|$ transformed view feature vectors for the current layer. Specifically, let $v_i^l$ be the feature vector of $i$-th view at the $l$-th layer.

The $v_i^l$ only depends on $v_i^{l-1}$ and can be obtained by the following mapping

$$v_i^l = M_i^l v_i^{l-1} \tag{8}$$

where $M_i^l$ is a mapping function for extracting more high-level feature representation $v_i^l$. This mapping function can be easily performed using a fully-connected layer followed by ReLU activation function.

**Feature fusion:** Fusion plays a very important role in multi-modal classification task. The PSR block is armed with two fusion operators. One is to fuse multiple view representation, and the other is to fuse the current fused feature representation and that of previous layer. It is well known that the quality of views usually differs. Similarly, the contribution of both of the previous fusion vector and current fusion vector may be different to final task. Based on above analysis, feature fusion is performed by the dual-weight fusion strategy, which takes $|V|$ transformed view feature vectors and the fusion vector from the previous layer, and outputs the fusion vector for the current layer. Specifically, this process of feature fusion can be divided into two steps for execution.

Firstly, it assigns weights to $|V|$ transformed view feature vectors based on their contribution to the classification task, and then cascades and transforms the weighted features into the fused transformed view feature. The cascading operation is able to not only unify multiple views together but also preserve their characters.

In our experiment, the weighted cascading function $f_v^l$ is designed by weighted concat of $|V|$ transformed view feature vectors $v_1^l, v_2^l, \cdots, v_{|V|}^l$ at the $l$-th layer, and can be obtained by following mapping

$$f_v^l = w_1^l v_1^l \oplus w_2^l v_2^l \cdots w_{|V|}^l v_{|V|}^l \tag{9}$$

where $\oplus$ is a cascading operator, $w_i^l$ is the automatically tuned weight parameter of $i$-th view at the $l$-th layer.

Next, the fused transformed view feature $v^l$ can be obtained by following mapping

$$v^l = M_v^l f_v^l \tag{10}$$

where $M_v^l$ is a mapping function transforms feature to the same dimension as the $v_i^l$, and it is performed using a fully-connected layer followed by ReLU activation function.

Secondly, it further adjusts the weights to the fused transformed view feature from the current layer and the fusion vector from the previous layer based on their contribution to the classification task, and then transforms the weighted features into the fused feature for the current layer. Specifically, let $c^{l-1}$ be the fusion vector at the ($l$-1)-th layer, and the $c_m^l$ is transformed from $c^l$ and can be obtained by following mapping.

$$c_m^l = M_m^l c^{l-1} \tag{11}$$

where $M_m^l$ is a mapping function for extracting more high-level feature representation $c_m^l$, just like $M_i^l$.

The fusion function $F^l$ is used to weighted fuse $v^l$ and $c_m^l$. In our experiment, the fusion operator is designed as *element-wise addition* or *outer product*, and it can be chosen one of them. The *element-wise addition* fusion operator is able to enhance every feature used for fusion, thereby strengthening the fused feature. The *outer product* operator can expand the dimension of the input vector to a higher-dimensional space, thus providing enhanced feature representation

capabilities. Afterwards, the output of fusion function $F^l$ is mapped through a mapping function to produce the final fused vector $c^l$. This step is implemented as follow:

$$c^l = M_c^l(w_{fv}^l v^l \otimes w_{fm}^l c_m^l) \tag{12}$$

where $w_{fv}^l$ and $w_{fm}^l$ are the two automatically tuned scale parameters, the $\otimes$ is the *element-wise addition* or *outer product* fusion operator, $M_c^l$ is a mapping function just like $M_i^l$ too.

Since *element-wise addition* and *outer product* in Eq. 12 are introduced, the magnitude of the output neurons $c^l$ may vary dramatically, and the model might converge to an unsatisfactory local minimum. In this paper, we use normalization [13] to address this problem. Specifically, we use the power normalization and $\ell_2$ normalization [7, 39]. This process is implemented as follows:

$$c^l \quad \leftarrow \quad \text{sign}(c^l)\sqrt{|c^l|} \tag{13}$$

$$c^l \quad \leftarrow \quad \frac{c^l}{\|c^l\|} \tag{14}$$

Compared to most existing fusion strategies that only fuse multiple view representation once via single fusion operator, the proposed PSR block can deal with two types of information stream: view representation and fused feature representation; Moreover, PSR can be simultaneously armed with two fusion operators, so it has greater flexibility for using exiting fusion operators. This reliefs the sub-optimal issue due to insufficient interaction among views.

## 3.2 Progressive Skip Reasoning Fusion Network for Multi-Modal Classification

We propose the progressive skip reasoning fusion network (PSRFN) by stacking multiple PSR blocks shown in Figure 2.

The first PSR block of the PSRFN network takes $|V|$ view features $v_1, v_2, \cdots, v_{|V|}$ and $c$ as inputs. $v_1, v_2, \cdots, v_{|V|}$ are extracted using different feature extractors such as SIFT and Hog. One simple native way is set input $c$ to $[0, 0, \ldots, 0]$.

**Skip:** The skip connection approach is adopted among PSR blocks, it is implemented by directly connecting the output of a PSR block to the input of subsequent PSR blocks, enabling efficient information flow between blocks. Specially, the output $c^1$ of first PSR block serves as the part of the input for the 2-th to the $L$-th PSR blocks, the output $c^2$ of sencond PSR block serves as the part of the input for the 3-th to the $L$-th PSR blocks, and so on in this manner.

Skip connections establish direct information channels between different layers, enabling the fusion of features from different layers, allowing the network to learn richer feature representations and more effectively utilize the feature information from each block during training. This contributes to improving the PSRFN's performance, convergence speed, and generalization ability in completing multi-modal classification tasks.

**Loss:** Ensemble learning, as an effective technique to integrate multiple models, tells us that the accuracy of each model is important to the final result [42]. Inspired this, different from most

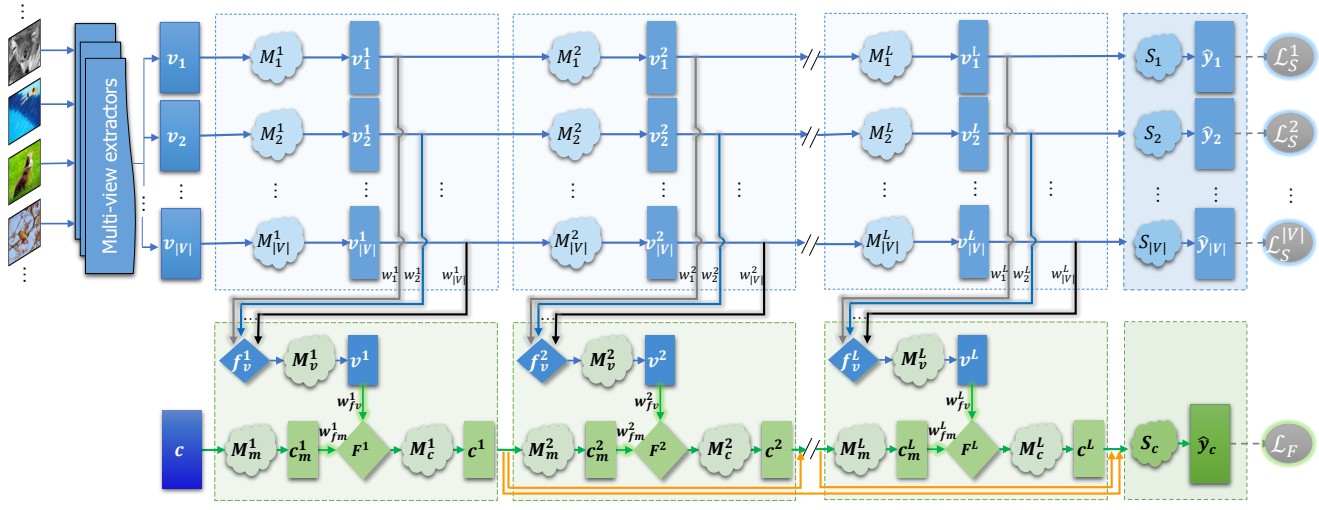

**Figure 2: The whole framework of the progressive skip reasoning fusion network (PSRFN)**

existing fusion frameworks which only optimize the fused information, we enforce loss function on each view representation and the final fused representation.

Given a set $D = (x, y)$ of $n$ examples where $x$ denotes any one example in $D$, its ground truth label is $y$. Its predicted probability for each class is $\hat{y}_j$ based on the $j$-th view data and it is obtained by

$$\hat{y}_j = \text{Softmax}(S_j v_j^L) \tag{15}$$

where $S_j$ is used to map $v_j^L$ into a special space whose dimensional is the same as the number of classes, and then the mapped vector is transferred a probability vector $\hat{y}_j$ with the Softmax function.

The total losses are defined as

$$\mathcal{L} = \mu \mathcal{L}_S + \lambda \mathcal{L}_F \tag{16}$$

where $\mathcal{L}_S$ is to ensure the accuracy of model taking each view and it is defined as

$$\mathcal{L}_S = \frac{1}{|V|} \sum_{j=1}^{|V|} \mathcal{L}_S^j = -\frac{1}{n|V|} \sum_{j=1}^{|V|} \sum_{(x,y) \in D} y \log(\hat{y}_j)$$

$\mathcal{L}_F$ is to ensure the accuracy of the fusion features and it is defined as

$$\mathcal{L}_F = -\frac{1}{n} \sum_{(x,y) \in D} y \log(\hat{y}_c)$$

$\mu$ and $\lambda$ are nonnegative tradeoff parameters controlling the relative contributions of the corresponding loss terms.

The flowchart of the entire PSRFN is illustrated in Fig. 2.

It should be noted that (1) the fusion operator used in this article can be replaced with other fusion operators such as bilinear pooling and tensor-based fusion methods; (2) *early fusion*, *intermediate fusion* and *late fusion* are three special cases of PSRFN.

## 4 EXPERIMENTAL STUDIES

In the experiments, we evaluate the effectiveness of the proposed PSRFN algorithm on six multi-modal classification datasets. Our computational environment is Ubuntu1 16.04.4, 512 GB DDR4 RDIMM,

2X 40-Core Intel(R) Xeon(R) CPU E5-2698 v4 @ 2.20GH and NVIDIA Tesla P100 with 16GB GPU memory.

### 4.1 Datasets

Our experiments are conducted on six challenging multi-modal classification datasets which include image, text, audio, depth and video datasets. (1) Animals with Attributes (AWA) [18] dataset, which includes 30,475 images from 50 categories with seven view features. These seven views consist of six low-level features and one deep feature. (2) NUS-WIDE-128 (NUS) [30] dataset, which includes 43,800 samples from 128 categories with seven view features. These seven views consist of six image features and one text feature. We select a subset of 10 categories from this dataset, with a total of 23,438 images. (3) Reuters [1] dataset, which includes 111,740 samples from six categories with five multilingual view features. To enable the model to process this dataset, PCA is used to reduce the dimensions of all views to 1000. According to [8, 24], Gaussian noise is added to either 5-view or 3-view datasets, resulting in two versions named Reuters5 and Reuters3, respectively. (4) VoxCeleb [27] dataset, which includes 153,516 samples from 1,251 categories with five audio view features. These five views consist of three traditional features and two deep features. To achieve the research objective, Gaussian noise is added to the two deep features. (5) YoutubeFace dataset, which includes 3,425 videos from 1,595 different people with five view features. According to [33], we use a subset of 31 categories from this dataset, with a total of 101,499 frames.

### 4.2 Evaluation Metric

We employ five measures to evaluate the performance of each method, which are accuracy, recall, precision, F1 score and kappa, respectively. Accuracy, recall and precision are widely-used measures. F1 is an indicator in statistics that comprehensively assesses the performance of the model by calculating the harmonic mean of

precision and recall. Wang et al. [31] pointed that the accuracy measure cannot recognize the random consistency and may cause an unreliable evaluation. Hence, one more reasonable measure: kappa is also used. These five evaluation metric definitions are as follows:

$$\text{accuracy} = \frac{TP + TN}{n} \tag{17}$$

$$\text{recall} = \frac{TP}{TP + FN} \tag{18}$$

$$\text{precision} = \frac{TP}{TP + FP} \tag{19}$$

$$\text{F1} = \frac{2 \times (precision \times recall)}{precision + recall} \tag{20}$$

$$\text{kappa} = \frac{p_o - p_e}{1 - p_e} \tag{21}$$

where $n$ denote the number of all samples; $TP$ and $TN$ denote the number of true positive and true negatives, respectively; $FP$ and $FN$ denote the number of false positives and false negatives, respectively. $p_o = \frac{TP+TN}{n}$ is the empirical probability of agreement on the label assigned to any sample (the observed agreement ratio), and $p_e = \frac{(TP+FN)\times(TP+FP)}{n} + \frac{(FP+TN)\times(FN+TN)}{n}$ is the expected agreement when both annotators assign labels randomly.

The larger values of the five metrics indicate a better classification performance. For each compared algorithm, 5-fold cross-validation is performed on each dataset, and the mean metric value and the standard deviation are recorded for comparative studies.

## 4.3 Experimental Results with Other Methods

PSRFN is a weighting-based multi-modal classification method (MMC). To validate the effectiveness of the proposed PSRFN, comprehensive comparison experiments are conduced with eight related weighting-based MMC methods. The compared methods can be classified into the following two groups according to the level of weighting:

(1) The first category is the *feature* level including EmbraceNet [5], AWDR [38] and RAMC [15]. EmbraceNet assigns 1 to the weight value of one view while 0 to others for each example according to a multinomial distribution. AWDR is an adaptive-weighting discriminative regression approach. Following [38], the parameter $\lambda$ is chosen from the set $\{10^{-3}, 10^{-2}, \cdots, 10^3\}$, while $k$ varies within the range $\{1, 3, \cdots, 9\}$. RAMC employs an $L_{2,1}$-norm loss function to acquire a joint weighted projection space across all views. This method preserves the correlation and diversity among views through a self-supervised weighting strategy. Similarly, the parameter $\lambda$ is chosen from $\{10^{-3}, 10^{-2}, \cdots, 10^3\}$, and $k$ ranges from $\{1, 3, \cdots, 9\}$.

(2) The second category is the *decision* level including BV [21], SSV [21], MR [21], TMC [8] and TMOA [25]. BV assigns 1 to the weight value of the view with the best performance while 0 to others according to whole classification performance of each view. MR assigns 1 to the weight value of the view with the best performance while 0 to others for each example according to the classification performance of each view of each example. SSV assigns the same values to all views. TMC and TMOA are two well-known trusted fusion

methods. Both of them model the confidence of each view at an evidence level with the Dempster-Shafer theory and opinion aggregation.

For the proposed PSRFN, in order to make the model more elegant and lightweight, we set each module to contain only one fully connected layer, and the number of neurons in the fully connected layer is selected from [64, 128, 256]. In light of comparison to other models, $\mu$ and $\lambda$ are fixed 1. PSRFN (*add.*) and PSRFN (*op.*) represent the use of *element-wise addition* fusion operator and *outer product* fusion operator, respectively, within the fusion function $F^l$.

The results, displayed in Tables 1 and 2, are presented through the mean metric value and the standard deviation obtained from 5-fold cross-validation. Avg. represents the average performance value of each algorithm across the six datasets, while Rank indicates the average ranking of each algorithm among all compared methods across the same datasets. From Tables 1 and 2, the following observations can be made:

(1) Compared to the BV method that can be viewed as single view method, PSRFN achieves the best classification performance, while other multi-modal fusion methods perform worse than BV on some datasets in terms of accuracy, recall, precision, F1 and kappa. For example, the accuracy of EmbraceNet and TMC is lower 1.11% and 10.83% than BV, respectively, while the proposed PSRFN is higher by 4.1% than BV on YoutubeFace dataset. The reason may be that PSRFN progressively enhances the fused features in a multiple interaction manner through adaptive combination with exiting fusion operators; while there is once interaction and no interaction for the multi-view features in EmbraceNet and TMC, respectively. This shows the advantage of our progressive fusion strategy.

(2) Compared to the multi-modal fusion methods, PSRFN statistically achieves the best results. Specifically, PSRFN wins 237 times out of 240 configurations (eight compared methods × six datasets × five metrics). It is noted that PSRFN achieves the comparable performance two times out of three loses, i.e., 87.73% vs 87.92% and 90.49% vs 90.64% (PSRFN vs other methods); PSRFN performs worse than the SSV method in terms of precision, but outperforms it in the other four evaluation metrics on YoutubeFace dataset. These results verify the effectiveness of PSRFN.

(3) For the five average classification metrics and Rank, the feature level weighting-based MMC methods statistically perform better than the decision level weighting-based MMC methods. For example, RAMC, which performs best among three feature level weighting-based MMC methods, achieves the average accuracy of 83.87%, whereas TMOA, the top performer among five decision level weighting-based MMC methods, attains the average accuracy of 82.36%. The reason for this may be that the former, which typically involves multiple view features, allows for more interaction among them than the latter. This further supports our viewpoint that the fused features may be enhanced through interacting with multi-view features more than once.

**Table 1: Comparison results (mean ± std) with SOTA algorithms on the accuracy, recall and precision, the best performance is highlighted in boldface.**

| | | | | | | | | | |
|---|---|---|---|---|---|---|---|---|---|
| **Accuracy** | | | | | | | | | |
| Groups | Methods | AWA | NUS | Reuters5 | Reuters3 | VoxCeleb | YoutubeFace | Avg. | Rank |
| Feature | EmbraceNet (IF19) | 84.97±0.23 | 72.43±0.38 | 80.07±0.21 | 83.58±0.25 | 81.74±0.34 | 80.90±1.04 | 80.62 | 7.33 |
| | AWDR (PR19) | 90.46±0.06 | 72.44±0.66 | 79.69±0.27 | 83.32±0.32 | 91.08±0.09 | 85.11±0.15 | 83.68 | 5.67 |
| | RAMC (INS22) | 90.63±0.13 | 72.51±0.67 | 79.84±0.25 | 83.48±0.25 | 91.54±0.11 | 85.21±0.17 | 83.87 | 4.50 |
| Decision | BV (TEVC21) | 88.65±0.43 | 68.69±0.59 | 80.61±0.25 | 83.98±0.14 | 63.25±0.14 | 82.01±0.18 | 77.87 | 7.00 |
| | SSV (TEVC21) | 82.37±1.26 | 63.70±0.64 | 79.51±0.41 | 84.71±0.22 | 85.10±0.23 | 84.43±0.31 | 79.97 | 7.00 |
| | MR (TEVC21) | 87.10±0.64 | 64.39±0.85 | 78.24±0.45 | 84.17±0.19 | 79.92±0.29 | 84.78±0.21 | 79.77 | 7.67 |
| | TMOA (AAAA22) | 89.17±0.31 | 72.60±0.48 | 79.11±0.43 | 84.19±0.27 | 84.72±0.21 | 84.35±0.25 | 82.36 | 6.00 |
| | TMC (TPAMI23) | 88.59±0.25 | 72.73±0.30 | 79.60±0.56 | 84.23±0.35 | 73.13±0.15 | 71.18±2.27 | 78.24 | 6.67 |
| Ours | **PSRFN (add.)** | **90.91±0.14** | 75.43±0.48 | 82.28±0.22 | 86.20±0.15 | **94.79±0.11** | **86.11±0.10** | 85.95 | 1.50 |
| | **PSRFN (op.)** | 90.49±0.17 | **75.49±0.38** | **82.36±0.17** | **86.23±0.17** | 93.65±0.11 | 86.03±0.35 | 85.71 | 1.67 |
| **Recall** | | | | | | | | | |
| Groups | Methods | AWA | NUS | Reuters5 | Reuters3 | VoxCeleb | YoutubeFace | Avg. | Rank |
| Feature | EmbraceNet (IF19) | 80.04±0.59 | 72.04±0.34 | 79.85±0.26 | 83.46±0.21 | 78.36±0.34 | 80.65±1.13 | 79.07 | 6.67 |
| | AWDR (PR19) | 86.86±0.20 | 71.87±0.62 | 79.59±0.23 | 83.30±0.29 | 87.26±0.13 | 83.57±0.30 | 82.08 | 5.50 |
| | RAMC (INS22) | **87.08±0.42** | 71.92±0.65 | 79.73±0.23 | 83.45±0.23 | 87.95±0.11 | 83.35±0.27 | 82.25 | 4.58 |
| Decision | BV (TEVC21) | 85.72±0.57 | 67.67±0.57 | 80.52±0.29 | 83.91±0.11 | 57.79±0.14 | 81.05±0.35 | 76.11 | 6.67 |
| | SSV (TEVC21) | 77.28±1.45 | 60.52±0.63 | 79.08±0.40 | 84.48±0.25 | 81.07±0.26 | 80.80±0.53 | 77.21 | 7.50 |
| | MR (TEVC21) | 83.55±0.77 | 63.10±0.91 | 78.11±0.45 | 84.12±0.26 | 75.36±0.32 | 83.87±0.31 | 78.02 | 7.33 |
| | TMOA (AAAA22) | 83.62±0.91 | 71.81±0.49 | 78.85±0.30 | 84.25±0.30 | 81.54±0.26 | 82.63±0.39 | 80.45 | 6.17 |
| | TMC (TPAMI23) | 84.47±0.54 | 71.70±0.43 | 79.60±0.56 | 84.19±0.29 | 64.06±0.12 | 68.50±2.77 | 75.42 | 7.17 |
| Ours | **PSRFN (add.)** | **87.08±0.35** | **74.93±0.43** | 82.11±0.17 | **86.19±0.22** | **93.26±0.14** | **85.46±0.39** | 84.84 | 1.25 |
| | **PSRFN (op.)** | 86.78±0.17 | 74.83±0.41 | **82.21±0.12** | 86.16±0.18 | 91.77±0.14 | 85.17±0.48 | 84.49 | 2.17 |
| **Precision** | | | | | | | | | |
| Groups | Methods | AWA | NUS | Reuters5 | Reuters3 | VoxCeleb | YoutubeFace | Avg. | Rank |
| Feature | EmbraceNet (IF19) | 82.14±0.57 | 71.73±0.32 | 80.42±0.25 | 83.77±0.34 | 80.95±0.46 | 83.71±1.10 | 80.45 | 7.50 |
| | AWDR (PR19) | 89.32±0.33 | 72.71±0.61 | 79.87±0.30 | 83.49±0.34 | 91.83±0.11 | 89.94±0.32 | 84.53 | 5.92 |
| | RAMC (INS22) | 89.41±0.38 | 72.82±0.64 | 80.12±0.27 | 83.70±0.28 | 92.19±0.06 | 90.64±0.08 | 84.81 | 4.17 |
| Decision | BV (TEVC21) | 86.57±0.46 | 70.98±0.95 | 80.77±0.19 | 84.13±0.19 | 64.63±0.63 | 84.34±0.61 | 78.57 | 7.17 |
| | SSV (TEVC21) | 82.76±1.10 | 67.23±0.58 | 80.19±0.49 | 85.16±0.20 | 84.44±0.18 | **94.13±0.37** | 82.32 | 5.33 |
| | MR (TEVC21) | 85.44±0.64 | 64.90±0.81 | 78.21±0.48 | 84.25±0.13 | 78.85±0.29 | 86.56±0.58 | 79.70 | 8.17 |
| | TMOA (AAAA22) | 88.15±0.62 | 72.73±0.53 | 79.89±0.72 | 84.40±0.23 | 84.38±0.30 | 87.59±0.28 | 82.86 | 5.50 |
| | TMC (TPAMI23) | 87.76±0.40 | 72.71±0.22 | 79.86±0.46 | 84.43±0.49 | 73.26±0.34 | 82.53±2.01 | 80.09 | 7.25 |
| Ours | **PSRFN (add.)** | **89.48±0.39** | 75.63±0.49 | 82.52±0.25 | 86.32±0.15 | **94.54±0.12** | 90.22±0.85 | 86.45 | 2.00 |
| | **PSRFN (op.)** | 88.77±0.43 | **75.82±0.44** | **82.61±0.20** | **86.38±0.20** | 93.29±0.09 | 90.49±0.97 | 86.23 | 2.00 |

In summary, the excellent performance of PSRFN indicates that it is highly competitive compared with other single fusion strategy-based methods.

## 4.4 Ablation Experiments

The PSRFN includes four components: $W_v$, $W_f$, skip and fusion operation. This subsection aims to show the role of each component of PSRFN through conducting ablation experiments on NUS and Reuters5 datasets. The results are shown in Table 3. The $W_v$ represents the component for weighting $|V|$ transformed view feature vectors $v_1^l, v_2^l, \cdots, v_{|V|}^l$ in the $l$-th PSRFN block. The $W_f$ represents the component for weighting the fused vector $v^l$ and $c_m^l$ in the $l$-th PSR block. The skip represents the use of skip connections between consecutive PSR blocks. Fusion operation represents the fusion of $v^l$ and $c_m^l$ using a fusion operator in the $l$-th PSR block. In this paper, the fusion operator can be either *add.* or *op.*, where *add.* represents *element-wise addition*, *op.* represents *outer product*. The settings are divided four groups according to the number that $W_v$, $W_f$ and skip are selected:

G1: None of the first three components is selected;
G2: One of the first three components is selected;
G3: Two of the first three components are selected;
G4: All of the first three components are selected.
From the Table 3, the following conclusions can be made.

(1) The best results are obtained by the PSRFN that is configured with $W_v$, $W_f$ and skip, which shows that each of three components plays an important role.

**Table 2: Comparison results (mean ± std) with SOTA algorithms on F1 and Kappa, the best performance is highlighted in boldface.**

| | | | F1 | | | | | | | |
|---|---|---|---|---|---|---|---|---|---|---|
| Groups | Methods | AWA | NUS | Reuters5 | Reuters3 | VoxCeleb | YoutubeFace | Avg. | Rank |
| Feature | EmbraceNet (IF19) | 80.60±0.62 | 71.78±0.36 | 80.07±0.22 | 83.59±0.25 | 78.64±0.41 | 81.61±0.99 | 79.38 | 7.33 |
| | AWDR (PR19) | 87.72±0.21 | 72.16±0.62 | 79.71±0.27 | 83.37±0.30 | 88.57±0.13 | 86.51±0.12 | 83.01 | 5.33 |
| | RAMC (INS22) | **87.92±0.30** | 72.21±0.65 | 79.90±0.25 | 83.54±0.24 | 89.20±0.10 | 86.69±0.17 | 83.24 | 4.00 |
| Decision | BV (TEVC21) | 85.94±0.50 | 68.64±0.63 | 80.61±0.24 | 83.99±0.11 | 58.34±0.23 | 82.49±0.25 | 76.67 | 6.83 |
| | SSV (TEVC21) | 78.82±1.42 | 62.13±0.69 | 79.49±0.42 | 84.75±0.21 | 81.75±0.23 | 86.55±0.27 | 78.92 | 6.83 |
| | MR (TEVC21) | 84.14±0.73 | 62.96±0.93 | 78.11±0.46 | 84.16±0.19 | 75.88±0.30 | 85.03±0.29 | 78.38 | 7.67 |
| | TMOA (AAAA22) | 83.65±0.86 | 72.02±0.49 | 79.14±0.49 | 84.24±0.24 | 82.02±0.33 | 84.85±0.25 | 80.99 | 6.33 |
| | TMC (TPAMI23) | 85.28±0.54 | 71.84±0.31 | 79.52±0.57 | 84.22±0.38 | 65.22±0.09 | 71.92±2.06 | 76.33 | 7.17 |
| Ours | **PSRFN (add.)** | 87.73±0.44 | 74.97±0.35 | 82.28±0.20 | 86.22±0.16 | **93.67±0.13** | **87.50±0.29** | 85.40 | 1.67 |
| | **PSRFN (op.)** | 87.38±0.25 | **75.11±0.37** | **82.37±0.14** | **86.25±0.18** | 92.22±0.12 | 87.49±0.36 | 85.14 | 1.83 |
| | | | Kappa | | | | | | | |
| Groups | Methods | AWA | NUS | Reuters5 | Reuters3 | VoxCeleb | YoutubeFace | Avg. | Rank |
| Feature | EmbraceNet (IF19) | 84.60±0.24 | 68.99±0.44 | 75.99±0.25 | 80.23±0.30 | 81.72±0.34 | 78.93±1.12 | 78.41 | 7.17 |
| | AWDR (PR19) | 90.23±0.06 | 68.96±0.75 | 75.54±0.32 | 79.91±0.38 | 91.07±0.09 | 83.40±0.18 | 81.52 | 5.83 |
| | RAMC (INS22) | 90.41±0.13 | 69.04±0.78 | 75.72±0.29 | 80.11±0.30 | 91.53±0.11 | 83.47±0.21 | 81.71 | 4.50 |
| Decision | BV (TEVC21) | 88.38±0.44 | 64.62±0.67 | 76.65±0.31 | 80.71±0.16 | 63.21±0.14 | 80.08±0.18 | 75.61 | 7.00 |
| | SSV (TEVC21) | 81.93±1.28 | 58.69±0.74 | 75.30±0.49 | 81.58±0.27 | 85.09±0.23 | 82.37±0.38 | 77.49 | 7.17 |
| | MR (TEVC21) | 86.78±0.65 | 59.69±0.98 | 73.79±0.54 | 80.93±0.24 | 79.89±0.29 | 83.16±0.22 | 77.37 | 7.67 |
| | TMOA (AAAA22) | 88.91±0.31 | 69.12±0.53 | 74.82±0.50 | 80.97±0.33 | 84.55±0.42 | 82.60±0.29 | 80.16 | 5.83 |
| | TMC (TPAMI23) | 88.31±0.26 | 69.22±0.35 | 75.44±0.68 | 81.01±0.42 | 73.10±0.15 | 67.68±2.46 | 75.79 | 6.67 |
| Ours | **PSRFN (add.)** | **90.69±0.14** | 72.35±0.52 | 78.65±0.26 | 83.38±0.19 | **94.79±0.11** | **84.58±0.10** | 84.07 | 1.50 |
| | **PSRFN (op.)** | 90.26±0.18 | **72.41±0.43** | **78.75±0.20** | **83.42±0.21** | 93.65±0.12 | 84.47±0.38 | 83.83 | 1.67 |

**Table 3: Ablation experiments, where × denotes that the corresponding component is removed, √denotes that the corresponding component is added.**

| Groups | ID | $W_v$ | $W_f$ | Skip | add. | op. | NUS | Reuters5 |
|---|---|---|---|---|---|---|---|---|
| G1 | 1 | × | × | × | √ | × | 75.08±0.44 | 81.89±0.20 |
| | 2 | × | × | × | × | √ | 74.86±0.42 | 81.94±0.17 |
| G2 | 3 | √ | × | × | √ | × | 75.16±0.43 | 82.13±0.20 |
| | 4 | √ | × | × | × | √ | 74.89±0.37 | 82.19±0.18 |
| | 5 | × | √ | × | √ | × | 75.18±0.30 | 81.98±0.19 |
| | 6 | × | √ | × | × | √ | 74.91±0.38 | 82.06±0.15 |
| | 7 | × | × | √ | √ | × | 75.12±0.46 | 81.90±0.23 |
| | 8 | × | × | √ | × | √ | 75.08±0.39 | 81.96±0.15 |
| G3 | 9 | √ | √ | × | √ | × | 75.25±0.50 | 82.16±0.18 |
| | 10 | √ | √ | × | × | √ | 74.92±0.45 | 82.22±0.16 |
| | 11 | √ | × | √ | √ | × | 75.18±0.47 | 82.24±0.14 |
| | 12 | √ | × | √ | × | √ | 75.09±0.34 | 82.25±0.18 |
| | 13 | × | √ | √ | √ | × | 75.32±0.43 | 82.10±0.16 |
| | 14 | × | √ | √ | × | √ | 75.35±0.38 | 82.07±0.16 |
| G4 | 15 | √ | √ | √ | √ | × | 75.43±0.48 | 82.28±0.22 |
| | 16 | √ | √ | √ | × | √ | **75.49±0.38** | **82.36±0.17** |

(2) When the configurations of the PSRFN change from G1 to G4 on two datasets, their accuracy values are 75.08%, 75.18%, 75.35%, 75.49% on NUS dataset, respectively; the accuracy values are 81.94%, 82.19%, 82.25%, 82.36% on Reuters5 dataset,

respectively. The consistent increase tend further implies that each component of PSRFN is very useful for the performance improvement.

In summary, the above evidence shows the design rationality of the PSRFN. Hence, it is worthwhile to pay more attention to the fusion operator usage and fused feature enhancement.

## 5 CONCLUSION

In this paper, we have proposed a progressive skip reasoning fusion network (PSRFN). Unlike most existing multi-modal fusion methods that only use one fusion operator in a single stage to fuse all view features, PSRFN utilizes a PSR block to fuse all views with the fusion operator at each layer. The comprehensive experimental results have verified the effectiveness of the proposed method, suggesting that the multiple interaction between fused feature and view features is beneficial. In the future, it is worthwhile to propose better progressive fusion strategy. It has been observed that PSRFN needs to be armed with different configurations. Hence, it is interesting to apply neural architecture search technique to automatically search for the proper configurations.

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
