# OpenReview forum: "A Progressive Skip Reasoning Fusion Method for Multi-Modal Classification"
_acmmm.org/ACMMM/2024/Conference — MM2024 Oral_

### Official Review · Reviewer_zc9E · 2024-05-06

**Rating:** 4
**Confidence:** 4

**Summary:**

A multi-modal fusion method is proposed based on the task of multi-modal learning, which combines feature-level and decision-level fusion.

**Strengths:**

##  **A simple and effective method.**
1. The idea is not complicated, and readers in the relevant field can easily understand it.
2. The idea of progressive skip reasoning fusion is interesting.
3. The motivation behind the method is clear, and the experiment is set based on the motivation, which complements each other.

**Limitations:**

## **Some minor concerns.**
This work provides readers with a simple and effective method, which currently appears to be a good work. But I still have some suggestions for the authors to improve the quality of this paper.

**In the related work section:** The fusion methods in Subsection 2.3 may not be fully summarized. Because there are other multi-modal fusion methods, such as weighted averaging [1], self-weighting [2], attention [3], and so on. If we can increase the discussion of these works, it will help improve the clarity of the contribution in this article.

[1] Tensorized multi-view subspace representation learning. IJCV, 2020.

[2] Dynamic auto-weighted multi-view co-clustering. PR 2020.

[3] GAF-Net: Graph attention fusion network for multi-view semi-supervised classification. ESWA 2023.

**In the method section:** The authors are suggested to add an algorithm flow chart in Subsection 3.2, which enhances the summarization of this work. Then, a complexity analysis of the model should be also added.

**In the experiment section:** The experiment was slightly incomplete due to the lack of sensitivity analysis for parameters $\mu$ and $\lambda$.

**Other details:** The overall layout and standardization still need to be improved for clear and good paper quality. For example, i) punctuation is required after formulas; ii) Some expressions need to be unified, such as Fig. 2 or Figure 2 in the main body.

**In summary, this is an easy to understand and effective multi-modal fusion work. Motivation, methods, and experiments are sufficient, but certain aspects can still be improved. If the authors are willing to adopt my suggestions to enhance the overall presentation quality of the paper, I am willing to increase my rating.**

**Suitability:**

3

---

### Official Review · Reviewer_3bZ7 · 2024-05-22

**Rating:** 4
**Confidence:** 3

**Summary:**

Authors have found that there is limited attention paid to the design of feature fusion usage in multimodal fusion studies. Therefore, the authors conducted research on how to better fuse features to enhance multimodal classification tasks. This paper proposed a progressive skip reasoning fusion network (PSRFN), which utilizes the progressive skip reasoning block to fuse all views with a fusion operator at each layer. This design enables the network to deeply understand and utilize multimodal information, leading to better performance in classification tasks. The PSRFN network provides novel ideas and methods for multimodal fusion research, demonstrating a certain degree of innovation and practicality.

**Strengths:**

This paper proposes a novel progressive fusion strategy that leverages the complementarity of multi-view information to gradually enhance the discriminative power of features. It also introduces a dual-weighted fusion strategy that precisely adjusts weights based on the actual contribution of each view feature and fusion layer feature, further improving the fusion effect. A progressive skip reasoning fusion network (PSRFN) for multi-modal classification is designed, eliminating the need for users to determine where the views should be fused. The network utilizes skip connections to fully capture both deep and shallow features, enhancing the model's ability to understand the data. Through extensive experimental validation, the authors proved that the design of feature fusion usage can indeed enhance performance, providing a direction for future research. The proposed PSRFN possesses a clear theoretical foundation, which reasonably explains the working principles and advantages of the model. It can also be extended to other multi-modal learning tasks, demonstrating broad application prospects.

**Limitations:**

1.The abstract should be modified according to the "motivation, description, results, and conclusion" sections. I suggest expanding the conclusion section to emphasize the significance of the findings.
2.The content in section 4.2 is relatively cumbersome. It needs to be reorganized to make it more concise and comprehensible.
3.The description of the model training process is unclear. It is necessary to explicitly state the stopping criteria for training, such as performance on a validation set, the number of iterations, or other specific conditions.

**Suitability:**

3

---

### Official Review · Reviewer_f5hX · 2024-05-24

**Rating:** 4
**Confidence:** 3

**Summary:**

This paper proposes a progressive skip reasoning fusion network (PSRFN) for multi-modal classification. Unlike most existing multi-modal fusion methods that only use one fusion operator in a single stage to fuse all view features, PSRFN utilizes a PSR block to fuse all views with the fusion operator at each layer. The comprehensive experimental results have verified the effectiveness of the proposed method.

**Strengths:**

This paper proposes a progressive skip reasoning fusion network (PSRFN) for multi-modal classification. This design method is novel and intersting.

**Limitations:**

Although this method is interesting, I still have some concerns that the author needs to explain:
1) Some detailed issues. All formulas have no punctuation marks, and through your symbols, it is not clear which are scalars, which are vectors, and which are tensors. Please rewrite.
2) Can your design idea be used in visual-language multi-modal classification tasks, such as multi-modal fake news detection, multi-modal sentiment analysis, multi-modal sarcasm detection, etc.? Please add some experiments to confirm your method.
3) Some symbols and letters in the frame diagram are not explained and I feel they are not rigorous enough. Please add them again.
4) How do you solve the problem of modal variability? If features conflict between multiple modalities, how does your design resolve them?
5) Your fusion seems quite simple, balanced through weight strategy, but I would like to ask how to explain the feature weight? You did not give some feature map visualization results in the experimental part. I want to intuitively see the feature visualization and fusion feature visualization of each single mode in the feature flow process, please add.
If the above issues are resolved, I will consider accepting it.

**Suitability:**

3

---

### Meta-Review · Area_Chair_pJU3 · 2024-06-26

**Recommendation:** Accept (Oral)
**Confidence:** 4

**Metareview:**

A well-written paper on skip reasoning. Technically strong as well.